# Performance of a Finnish Diabetes Risk Score in detecting undiagnosed diabetes among Kenyans aged 18–69 years

Innocent B. Mugume[1,2]*, Solomon T. Wafula[3�], Damazo T. Kadengye[4ø‡], Josefien Van Olmen[5ø‡]

1 Department of Integrated Epidemiology, Surveillance and Public Health Emergencies, Ministry of Health, Kampala, Uganda, 2 Department of Epidemiology and Social Medicine, Faculty of Medicine and Health Sciences University of Antwerp, Antwerp, Belgium, 3 Department of Disease Control and Environmental Health, School of Public Health, Uganda Makerere University, Kampala, Uganda, 4 African Population and Health Research Center, Nairobi, Kenya, 5 Department of Family Medicine and Population Health, Global Health Institute, University of Antwerp, Antwerp, Belgium

ø These authors contributed equally to this work.
‡ DTK and JVO are joint senior authors on this work.
* ibmugume@gmail.com

**Data Availability Statement:** All relevant data are within the paper and its Supporting information files.

**Funding:** The author(s) received no specific funding for this work.

## Abstract

### Background

The application of risk scores has often effectively predicted undiagnosed type 2 diabetes in a non-invasive way to guide early clinical management. The capacity for diagnosing diabetes in developing countries including Kenya is limited. Screening tools to identify those at risk and thus target the use of limited resources could be helpful, but these are not validated for use in these settings. We, therefore, aimed to measure the performance of the Finnish diabetes risk score (FINDRISC) as a screening tool to detect undiagnosed diabetes among Kenyan adults.

### Methods

A nationwide cross-sectional survey on non-communicable disease risk factors was conducted among Kenyan adults between April and June 2015. Diabetes mellitus was defined as fasting capillary whole blood $\geq$ 7.0mmol/l. The performance of the original, modified, and simplified FINDRISC tools in predicting undiagnosed diabetes was assessed using the area under the receiver operating curve (AU-ROC). Non-parametric analyses of the AU-ROC, Sensitivity (Se), and Specificity (Sp) of FINDRISC tools were determined.

### Results

A total of 4,027 data observations of individuals aged 18−69 years were analyzed. The proportion/prevalence of undiagnosed diabetes and prediabetes was 1.8% [1.3–2.6], and 2.6% [1.9–3.4] respectively. The AU-ROC of the modified FINDRISC and simplified FINDRISC in detecting undiagnosed diabetes were 0.7481 and 0.7486 respectively, with no statistically significant difference (p = 0.912). With an optimal cut-off $\geq$ 7, the simplified FINDRISC had a

**Competing interests:** The authors have declared that no competing interests exist.

higher positive predictive value (PPV) (7.9%) and diagnostic odds (OR:6.65, 95%CI: 4.43–9.96) of detecting undiagnosed diabetes than the modified FINDRISC.

## Conclusion

The simple, non-invasive modified, and simplified FINDRISC tools performed well in detecting undiagnosed diabetes and may be useful in the Kenyan population and other similar population settings. For resource-constrained settings like the Kenyan settings, the simplified FINDRISC is preferred.

## Introduction

Globally, diabetes mellitus, a chronic and metabolic disease characterized by increased levels of blood glucose (or blood sugar) resulting from defects in insulin production, ineffective body use of the insulin produced, or both [1, 2], is growing to epidemic proportions owing to rising levels of obesity, physical inactivity and inappropriate diet [3]. In 2019, global estimates of diabetes among 20-79-year-old adults were 463 million and were projected to rise beyond 700 million by 2045 [4]. Type 2 diabetes (T2D) is the most common, accounting for almost 90% of all diabetes cases worldwide, and results mainly from a confluence of environmental, behavioral, and/or genetic factors [4]. T2D is often asymptomatic in its earliest stages, with many cases remaining undiagnosed until the clinical manifestation of related complications, especially at the micro-and macro-vascular levels [4–6]. Africa has the highest proportion of undiagnosed diabetes (about 60%) compared to the other regions in the world [4, 7].

In Kenya, the proportion of undiagnosed is estimated at 53% [7, 8], almost the same as that of the continental average.

Undiagnosed diabetes is a major risk factor for premature death and other severe health complications, such as blindness, cardiovascular complications, and peripheral artery disease, among others [9–11]. Moreover, the economic burden of treating diabetes and its related complications is likewise enormous, and it affects households' income and savings, propagating the spiral of ill health and poverty. The loss of productivity and healthcare costs incurred also threatens national economies [12, 13]. However, studies have demonstrated that effective prevention and early detection delays the onset of diabetes-related complications, and prevent associated premature deaths [14, 15]. Current diabetes diagnosis and/or targeted screening strategies such as glycated hemoglobin, fasting plasma glucose, or 2-hourly random glucose measurement and point of care capillary blood glucose measurement recommended for use in low and middle-income countries (LMICs) [16], are invasive, inconvenient, and not feasible for population-wide screening [17, 18].

Additionally, most primary care systems in LMICs do not have sufficient human resources and diagnostic capacity to scale up recommended diabetes diagnosis and/or screening strategies at all levels of the health care system [19]. The use of diabetes risk scores to be used for active community-based or facility-based screening for people at risk has thus been advocated for, to identify high-risk asymptomatic individuals to be further tested. Among the many diabetes risk score tools developed from different populations to identify individuals at high risk of developing T2D [20–25], is the Finnish Diabetes Risk Score (FINDRISC). It is the most widely used risk score owing to its validity and ease of applicability.

Although, the FINDRISC tool was originally developed, validated and used as a screening tool to predict risk of developing type 2 diabetes across Europe [26], and in some LMICs [27,

28], it has also been used for predicting current diabetes, but there is no evidence of its validation and use in Kenya. Population-based diabetes screening by measuring fasting blood glucose is not only a costly and invasive technique, but is also not practical and, not readily available in the community and primary care settings in some LMICs including Kenya [29, 30]. This study, therefore, aimed to measure the performance of the FINDRISC in detecting undiagnosed diabetes among adults (18–69) years in Kenya.

## Materials and methods

### Study design

Analyses were conducted using data from the Kenya STEPwise survey, a population-based cross-sectional household survey conducted between April and June 2015, and designed to provide estimates for indicators on non-communicable diseases (NCDs) risk factors among adults in Kenya aged 18–69 years. The study design, sampling, and implementation of the Kenya STEPwise survey have been described in detail elsewhere [31].

### Study population and sampling

Eligible participants were those aged between 18 and 69 years who had lived in the selected household for at least six months. Women who reported being pregnant, or individuals who were unable to stand or communicate, or refused to participate in the survey were excluded.

A multi-stage stratified sampling design involving a selection of clusters, households, and eligible individuals was used to select a representative sample of the general adult population of Kenya. The sample size was estimated using a standard formula based on a 50% prevalence of risk factors as no representative figures were available for the Kenya population at that time. A three-stage cluster sample design was used; in the first stage, 200 clusters (100 urban and 100 rural) were selected. In the second stage, a uniform sample of 30 households was selected from the listed households in each cluster, while in the third stage, one individual was randomly selected from all eligible listed household members. With a target of 30 households from each cluster, a minimum sample size of 6,000 individuals was derived to allow for national estimates as per sex and residence (rural or urban).

To produce unbiased estimates, sampling weights derived from the creation of a fifth National Sample Survey and Evaluation Programme (NASSEP V) master sampling frame from the Kenya National Bureau of Statistics, developed using the enumeration areas generated from the 2009 Kenyan Population and Housing Census was used. Participant selection processes were calculated as the inverse of all selection probabilities.

### Data collection

The data used in this study were collected in three steps. All data collectors were trained on the study protocol, human subject protection and supervised during actual data collection. The Step 1 data included: Socio-demographics and self-reports of NCD behavioral risk factors such as tobacco use, past medical histories of raised blood pressure and diabetes mellitus, and physical activity (PA). PA was assessed using the WHO Global Physical Activity Questionnaire [32] which was incorporated in the main study questionnaire.

Step 2 data comprised of physical measurements: standing height, weight, blood pressure, waist, and hip circumferences. Weight was measured (to the nearest 0.1 kgs) using a pre-calibrated digital weighing scale (SECA® 877), while the participant stood barefoot with light clothing and standing height to the nearest 0.5 cm using a portable stadiometer (SECA® 877).

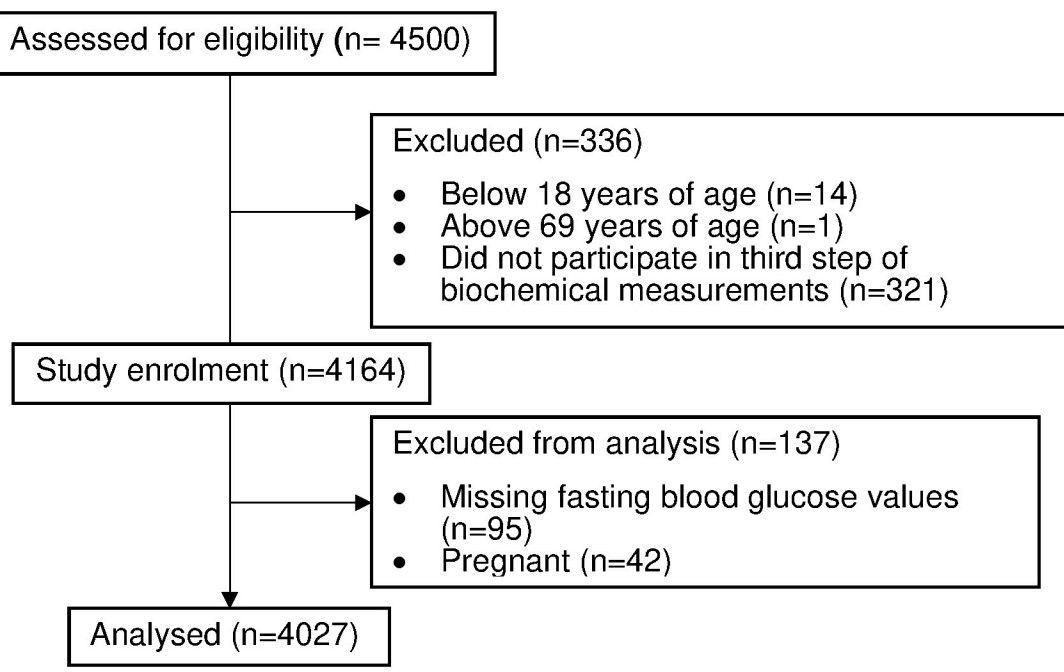

**Fig 1. Study participant enrolment flow chat.**

Waist circumference (to the nearest 0.5 cm) was measured using a constant tension standard tape measure and measurement was taken midway between the lowest rib and iliac crest. Blood pressure measurement was taken with the participant in a sitting position using a validated digital blood pressure machine (OMRON M2 device). Three readings were taken 3–5 minutes apart for both systolic and diastolic pressure following WHO guidelines [33].

Step 3 data included biochemical measurements taken after overnight fasting and included fasting blood glucose (FBG). Finger-prick venous blood samples were collected and analyzed using a Point of Care CardioCheck® PA analyzer. Of the 4500 eligible participants, 4164 participants were enrolled for this study procedures while 336 participants were excluded (14 were below 18 years, 1 above 69 years, and 321 did not participant in the third step of biochemical measurements). For this study analysis, 137 participants were further excluded (95 had missing fasting glucose values and 42 were pregnant) leaving a final data sample size of 4027 for analysis (Fig 1).

## Definitions

For this study, impaired fasting glycemia (IFG) was defined as fasting capillary whole blood $\geq$ 6.1 mmol/l but less than 7.0 mmol/l while diabetes mellitus was defined as fasting capillary whole blood $\geq$ 7.0 mmol/l [2, 34, 35]. Hypertension was defined as systolic blood pressure $\geq$ 140 mmHg and diastolic blood pressure $\geq$ 90 mmHg or previous diagnosis of hypertension or taking medication for raised blood pressure [33]. Body Mass Index (BMI) was computed by dividing the weight (kgs) by the height in meters squared ($m^2$) and used to develop categories of underweight, normal weight, overweight, and obesity [36]. Abdominal obesity was measured as waist circumference $\geq$ 94 cm for men and $\geq$ 80 cm for women [37]. PA was expressed as self-report of minutes of moderate-intense or vigorous-intense physical activity per week and derived from the WHO Global Physical Activity Questionnaire [32].

Insufficient physical activity in this study was defined as self-report of less than 150 minutes of moderate-intense or less than 75 minutes of vigorous-intense physical activity per week, including walking, running, and cycling. Fruit and vegetable consumption was assessed based on self-reports of fruit and vegetable servings per day. Insufficient fruit and vegetable consumption was defined as self-reports of less than 5 fruit and vegetable servings per day in a typical week.

## Original finnish (FINDRISK) diabetes risk score

The FINDRISC is a simple diabetes risk score tool originally developed in Finland to predict incident diabetes among the Finnish population aged 35–64 years [22]. It's based on eight simple diabetes risk factors including age (years), BMI (kgs/m$^2$), waist circumference (cm), history of high blood pressure, history of raised blood sugar, family history of diabetes, daily consumption of fruits or vegetables and daily physical activity [22]. The tool does not require laboratory investigations, and has different weighted scores according to the associated risk, with the final score ranging from 0 to 26 [26].

## Statistical analyses

All statistical analyses were done using Stata version 16.0 (Stata Corp, College Station, Texas, USA). All data for this analysis were weighted based on the primary data survey weights. Continuous variables were presented as median (IQR), and categorical variables as absolute and weighted frequencies. The prevalence and 95% confidence interval (95% CI) of IFG and undiagnosed T2D were calculated. Weighted logistic regression analyses were performed and because data were not normally distributed and contained outliers, the non-parametric Receiver Operating Characteristic (ROC) curve of the modified and simplified FINDRISC scores was constructed to demonstrate the true positive and false positive relationships and AUC determined. Sensitivity, specificity, diagnostic accuracy, positive predictive values (PPV), negate predictive values (NPV), positive and negative likelihood ratios (LHR), and diagnostic odds ratio (OR) of the risk score tools were also determined. The optimal cut-off points were determined as the points with the shortest distance in the ROC curve and, calculated as the square root of $[(1- \text{sensitivity})^2 + (1- \text{specificity})^2]$ [38].

## Creating a modified FINDRISC

Modifications to the original FINDRISC tool were made based on available secondary analysis data. The primary study data collection instrument had no question on parental/family history of diabetes thus, data to that specific risk factor was not collected. However, data to the other FINDRISC tool variables: age, BMI, waist circumference, physical activity, fruit and/or vegetable consumption, personal history of hypertension and history of high blood glucose were collected, and are what were included in the logistic regression analysis. The same original FINDRISC score for the different risk factor components was maintained. With this modification, the maximum score of the modified FINDRISC thus reduced from 26 of the original FINDRISC to 20. The diagnostic accuracy of the modified FINDRISC was assessed using the area under the ROC curve as well as sensitivity and specificity.

## Creating a simplified FINDRISC

Finally, modified FINDRISC was further simplified including, only variables independently associated with undiagnosed diabetes in our sample. Hence fruit and/or vegetable consumption, and physical activity variables did not significantly influence the AUC in detecting

undiagnosed diabetes and were excluded. Only age, BMI, waist circumference, and history of high blood glucose and history of hypertension were retained, creating a simplified FINDRISC, with a maximum score of 18. Diagnostic accuracy, sensitivity, and specificity were also assessed.

## Ethical considerations

The study protocol was designed in compliance with the Helsinki declaration and approved by the Kenya Medical Research Institute (KEMRI) Ethics Committee. All participants gave informed consent to study procedures before primary data collection. For this analysis, protocol approval was obtained from the University of Antwerp Ethics Committee and World Health Organization (WHO) NCD microdata repository [39], and access to data use was provided by the WHO STEPS team data repository administrator. The present report is presented according to the Strengthening the Reporting of Observational Studies in Epidemiology (STROBE) [40]

## Results

### Characteristic of the study population

Of the 4164 participants who were enrolled in the study, a total of 4027 participants' data were considered in this analysis while 137 were excluded (95 had missing fasting glucose values and 42 were pregnant as shown in Fig 1. About 50% of the study participant were males aged 18–69 years (median age: 36 [IQR:27–47]). Detailed characteristics of participants are shown in Table 1.

### Prevalence of undiagnosed diabetes and prediabetes

The weighted prevalence of undiagnosed diabetes was 1.8% [CI:1.3–2.6] and that of prediabetes was 2.6% [CI:1.9–3.4]. According to gender, women had a higher weighted proportion/prevalence of undiagnosed diabetes (2.4% [CI: 1.7–3.3]) and prediabetes (2.8% [CI: 2.1–3.8]) than was noted among men (1.3% [CI: 0.7–2.3]) and (2.3% [CI: 1.5–3.4]) respectively (Table 1).

### Performance of the modified FINDRISC

Overall, the modified FINDRISC scores ranged from 0 to 20, with a median of 3 (IQR:1–6). Participants with undiagnosed diabetes had a score range of 0 to 18, with a median of 9 (IQR:3–12). The median score and IQR differed for both undiagnosed diabetes and prediabetes differed across gender. When assessing the diagnostic accuracy of modified FINDRISC in detecting undiagnosed diabetes and prediabetes, the AU-ROC was 0.748 (CI: 0.692–0.804) and 0.631 (CI: 0.576–0.685) respectively (Table 2).

### Simplification of FINDRISC for the Kenyan population

With the simplified FINDRISC, scores ranged from 0 to 18, with a median of 2 (IQR:0–5). Participants with undiagnosed diabetes and prediabetes identified by simplified FINDRISC had a median of 8 (IQR:3–11) and 5 (IQR:1–8) respectively. The AU-ROC of the simplified FINDRISC in detecting undiagnosed diabetes and prediabetes was 0.749 (CI: 0.692–0.805) and 0.636 (CI: 0.583–0.688) respectively. The median score and IQR differed for both undiagnosed T2D and prediabetes across gender (Table 2). However, the diagnostic accuracy of the simplified FINDRISC was similar to the modified FINDRISC in detecting undiagnosed diabetes (P = 0.912) (Fig 2)

**Table 1. Characteristics of study participants, according to normal fasting glucose, impaired fasting glucose, and diabetes.**

| | Missing (%) | Gender (M/F) | All, n = 4,027 | Normoglycemia, n (weighted %) 3,805 (95.6) | Elevated fasting blood glucose | |
| --- | --- | --- | --- | --- | --- | --- |
| | | | | | Prediabetes n (weighted %) 123 (2.6) | Diabetes n (weighted %) 99 (1.8) |
| **Proportion/ Prevalence [95% CI]** | | | | 95.6% | 2.6% | 1.8% |
| | | | | [94.4–96.5] | [1.9–34] | [1.3–2.6] |
| **Gender distribution, Proportion/ Prevalence [95% CI]** | | M | 50.6% | 96.4% | 2.3% | 1.3% |
| | | | [47.8–53.4] | [95.0–97.4] | [1.5–3.4] | [0.7–2.3] |
| | | F | 49.4% | 94.7% | 2.8% | 2.4% |
| | | | [46.6–52.2] | [93.3–95.9] | [2.1–3.8] | [1.7–3.3] |
| **Age (years)** | | M | 36 (27–47) | 35 (27–46) | 40 (30–56) | 44 (35–55) |
| | | F | 35 (27–47) | 35 (27–46) | 37 (29–58) | 51 (40–60) |
| **BMI (kgs/m$^2$)** | 14 | M | 21.6 | 21.6 | 21.1 | 23.0 |
| | (0.35) | | (19.5–24.0) | (19.5–23.9) | (19.1–26.2) | (18.5–26.8) |
| | | F | 23.5 | 23.3 | 24.2 | 27.9 |
| | | | (20.4–27.5) | (20.3–27.2) | (21.4–29.4) | (23.7–30.5) |
| **Waist circumference (cm)** | 8 (0.2) | M | 79.5 | 79.0 | 85.0 | 90.0 |
| | | | (72.0–89.0) | (71.5–88.0) | (76.0–94.0) | (75.0–99.0) |
| | | F | 78.0 | 78.0 | 81.3 | 81.0 |
| | | | (72.0–85.0) | (72.0–85.0) | (74.0–91.0) | (71.5–95.0) |
| **Systolic BP (mmHg)** | 2(0.05) | M | 125.0 | 125.0 | 129.5 | 129.0 |
| | | | (116.5–135.5) | (116.5–135.0) | (120.0–151.5) | (112.5–150.0) |
| | | F | 120.5 | 120.0 | 121.6 | 132.5 |
| | | | (111.5–132.0) | (111.5–131.5) | (109.5–138.5) | (118.5–152.0) |
| **Diastolic BP (mmHg)** | 2(0.05) | M | 81.0 | 81.0 | 82.0 | 84.8 |
| | | | (73.5–88.5) | (73.5–88.0) | (73.0–93.5) | (73.0–100.0) |
| | | F | 81.5 | 81.5 | 82.0 | 88.0 |
| | | | (74.5–89.0) | (74.0–88.5) | (74.6–90.0) | (79.0–95.5) |
| **Physical inactivity (<30 minutes/ day)** | | M | 55 (1.9) | 53 (3. 5) | 0 (0) | 2 (0.3) |
| | | F | 114 (1.6) | 100 (2.8) | 8 (0.2) | 6 (0.1) |
| **Daily Fruit and vegetable consumption (No)** | | M | 891 (1.9) | 851 (5.6) | 23 (1.1) | 17 (0.8) |
| | | F | 1,358 (1.6) | 1,280 (5.5) | 49 (1.8) | 29 (1.3) |
| **History of High BP (mmHg) or on medication** | | M | 109 (2.8) | 97 (4.8) | 6 (0.3) | 6 (0.4) |
| | | F | 335 (5.9) | 295 (10.8) | 13 (0.3) | 27 (0.9) |
| **History of High blood glucose** | | M | 21 (2.8) | 14 (0.7) | 2 (0.2) | 5 (0.1) |
| | | F | 48 (5.9) | 24 (0.6) | 4 (0.1) | 20 (0.8) |

Data are presented as median, IQR: Inter Quartile Range; and n (weighted %) unless otherwise specified; n: count number; CI: Confidence Interval.

Table 3, shows the sensitivity, specificity, diagnostic accuracy, PPV, NPV, LHRs, and the diagnostic OR of six different cut-off points of the modified and simplified FINDRISC if used as tools for screening undiagnosed diabetes. With a cut-off of ≥ 7, the modified FINDRISC had a 2% higher sensitivity but with a 3.3% lower specificity level than the simplified FINDRISC. The diagnostic accuracy is the same (0.70) but the simplified FINDRISC had a higher PPV (7.9%) and diagnostic odds (OR:6.65, CI: 4.43–9.96) of detecting undiagnosed diabetes than the modified FINDRISC, S1 Table.

**Table 2. Modified and simplified finnish risk score according to normal fasting glucose, impaired fasting glucose, and diabetes.**

| | | All, n = 4,027 | Normoglycemia, n (weighted %) 3,805 (95.6) | Elevated fasting blood glucose | |
|---|---|---|---|---|---|
| | | | | Prediabetes n (weighted %) 123 (2.6) | Diabetes n (weighted %) 99 (1.8) |
| Modified Finnish Diabetes Risk Score, by gender | Median score [IQR] | 3 [1–6] | 3 [1–6] | 5 [1–8] | 9 [3–12] |
| | Male | 2 [1–4] | 2 [1–4] | 3 [1–7] | 5.5 [2–8] |
| | Female | 4 [1–7] | 4 [1–7] | 7 [3–9] | 10 [5–13] |
| | Range | 0–20 | 0–19 | 0–18 | 0–18 |
| | ROC-AUC [95% CI] | | | 0.631 | 0.748 |
| | | | | [0.576–0.685] | [0.692–0.804] |
| Simplified Finnish Diabetes Risk Score, by gender | Median score [IQR] | 2 [0–5] | 2 [0–5] | 5 [1–8] | 8 [3–11] |
| | Male | 1 [0–3] | 1 [0–3] | 2 [0–6] | 4.5 [1–8] |
| | Female | 3 [0–7] | 3 [0–6] | 6 [3–8] | 9 [5–12] |
| | Range | 0–18 | 0–18 | 0–17 | 0–18 |
| | ROC-AUC [95% CI] | | | 0.636 | 0.749 |
| | | | | [0.583–0.688] | [0.692–0.805] |

95% CI: Confidence Interval; n: counts; ROC-AUC: area under the receiver operating characteristics curve.

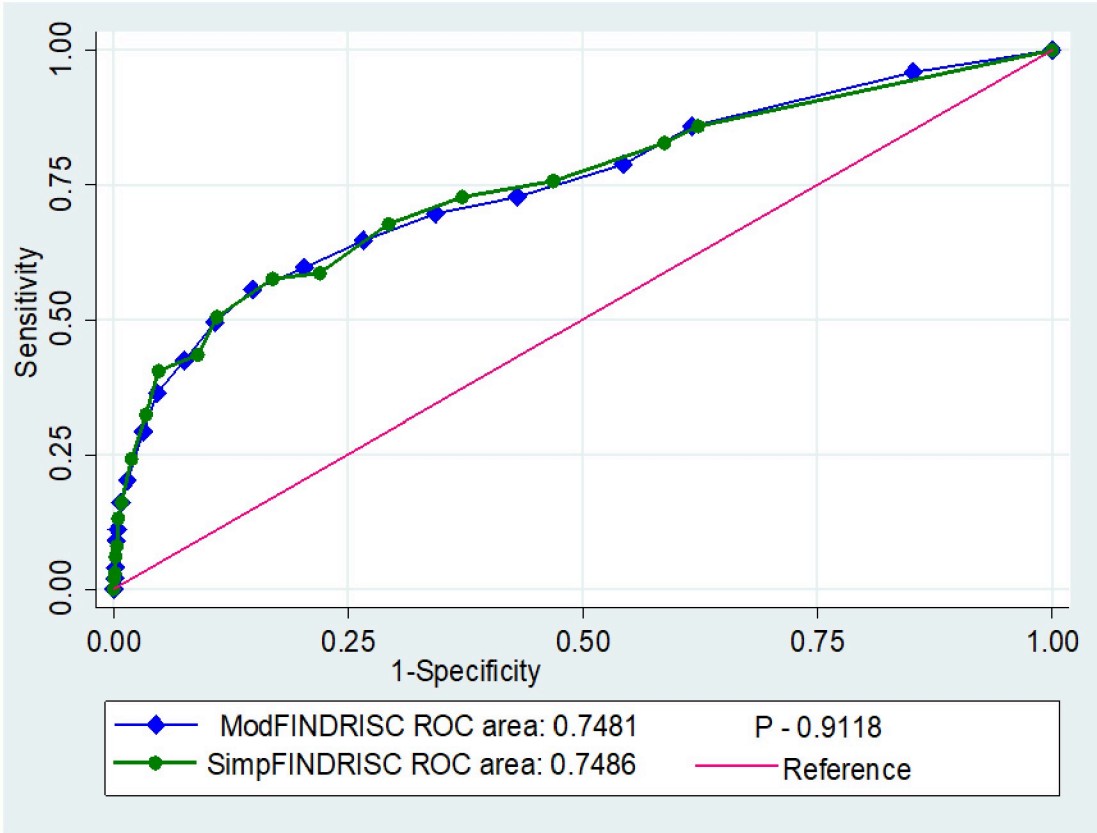

**Fig 2. ROC comparison between modified FINDRISC and simplified FINDRISC.**

**Table 3. Sensitivity, specificity, and predictive values of different cut-off points of the modFINDRISC and simpFINDRISC in the diagnosis of undiagnosed diabetes.**

| Cut-off | N, Population Identified (% tot pop) | Sensitivity (%) | Specificity (%) | Diagnostic accuracy | PPV (%) | NPV (%) | (+) LHR | (−) LHR | Diagnostic Odds | |
|---|---|---|---|---|---|---|---|---|---|---|
| | | | | | | | | | OR | 95% CI |
| Modified FINDRISC | | | | | | | | | | |
| ≥ 4 | 1,764 (43.8) | 72.7 | 56.9 | 0.65 | 4.1 | 98.8 | 1.69 | 0.35 | 3.52 | 2.26–5.49 |
| ≥ 5 | 1,417 (35.2) | 69.7 | 65.7 | 0.68 | 4.9 | 98.9 | 2.03 | 0.46 | 4.40 | 2.86–6.77 |
| ≥ 6 | 1,110 (27.6) | 64.6 | 73.4 | 0.69 | 5.8 | 98.8 | 2.43 | 0.48 | 5.04 | 3.33–7.63 |
| **≥ 7** | **858 (21.3)** | **59.6** | **79.7** | **0.70** | **6.9** | **98.7** | **2.93** | **0.51** | **5.78** | **3.85–8.67** |
| ≥ 8 | 638 (15.8) | 55.6 | 85.2 | 0.70 | 8.6 | 98.7 | 3.74 | 0.52 | 7.17 | 4.79–10.74 |
| ≥ 9 | 472 (11.7) | 49.5 | 89.2 | 0.69 | 10.9 | 98.6 | 4.60 | 0.57 | 8.12 | 5.42–12.17 |
| Simplified FINDRISC | | | | | | | | | | |
| ≥ 4 | 1,531 (38.0) | 72.7 | 62.9 | 0.68 | 4.7 | 98.9 | 1.96 | 0.43 | 4.51 | 2.90–7.03 |
| ≥ 5 | 1,219 (30.3) | 67.7 | 70.7 | 0.69 | 5.5 | 98.9 | 2.31 | 0.46 | 5.05 | 3.30–7.71 |
| ≥ 6 | 920 (22.8) | 58.6 | 78.1 | 0.68 | 6.3 | 98.7 | 2.67 | 0.53 | 5.03 | 3.36–7.54 |
| **≥ 7** | **723 (18.0)** | **57.6** | **83.0** | **0.70** | **7.9** | **98.7** | **3.40** | **0.51** | **6.65** | **4.43–9.96** |
| ≥ 8 | 484 (12.0) | 50.5 | 88.0 | 0.70 | 10.4 | 98.6 | 4.57 | 0.56 | 8.21 | 5.48–12.31 |
| ≥ 9 | 396 (9.8) | 43.4 | 91.0 | 0.67 | 10.9 | 98.5 | 4.83 | 0.62 | 7.78 | 5.16–11.72 |

## Discussion

The FINDRISC tool has largely been validated in Caucasian populations and is widely recommended as a simple diabetes screening tool across Europe and in other various population settings [26, 41–43] but has to a less extent been validated in the African settings [44]. Using data from a large national cross-sectional survey, this study analysis showed that the AU-ROC of the modified and simplified FINDRISC were 0.748 and 0.749 respectively. This performance was lower than the AUC of 0.87 obtained in the first validation study among the Finnish population [22] but better than some other validation studies conducted in Finland with an AUC of 0.727 [26]; Bulgaria with an AUC of 0.70 [42] and Greece with AUC of 0.724 [41]. More so, the performance was better than the findings from a large primary care study conducted in Spain with an AUC of 0.69 [45]. Whereas this study sample size was limited before this analysis was undertaken, it is still very comparable to those in other studies such as the original study in Finland and the one in Philippines, and its findings were generally similar to other several validation studies conducted in diverse settings, with AUC ranging between 0.65 and 0.88 [41, 44, 46]. As such, the FINDRISC can be adopted and utilized in a community screening of T2D in Kenya and other similar contexts.

Although, there are insufficient data on prior FINDRISC use in the African settings, a study from South Africa that assessed the performance of five different risk score tools in predicting undiagnosed diabetes in a diverse racial population of Cape Town, had comparable AUC findings of identifying undiagnosed diabetes [47]. Similarly, findings from another cross-sectional study conducted in Bostwana had comparable but slightly lower AUC of 0.63 [48]. Additionally, findings from a study conducted in Algeria to identify individuals with dysglycemia using FINDRISC were similarly lower, with an AUC of 0.64 [49]. Despite the low performance of the FINDRISC documented in a few studies conducted in some Africa countries, this study's overall findings of AUC of 0.748 is better than in other contexts and thus useful in detecting people who might have undiagnosed diabetes and should receive a diagnostic test [50].

The observed differences could be in part, attributable to diverse population characteristics and the genetic variants inherent in the African population, and the possible use of different diabetes diagnostic tests such as glycated hemoglobin, FBG, or Oral glucose tolerance test (OGGT) that have varying sensitivities and specificities in different settings [51, 52].

At univariate and multivariate analyses, physical activity and fruit or vegetable consumption variables were excluded because they were not associated with undiagnosed diabetes. Additionally, a weighted proportion of 1.6% of those who reported engaging in daily physical activity for at least 30 minutes had undiagnosed diabetes, and 0.8% of those who reported daily consumption of fruits and/or vegetables too had undiagnosed diabetes. With these adjustments, there was no statistically significant difference (P = 0.912) between the simplified FINDRISC and the modified FINDRISC in detecting individuals with undiagnosed diabetes.

To maximize true positive rates and minimize false-negative rates, the optimal cut-off score of the FINDRISC was selected based on the trade-off between sensitivity and specificity, diagnostic accuracy, predictive values, and diagnostic odds. The sensitivity and the specificity of any screening test, and also the prevalence of the disease in the population being screened determine the predictive values of a screening test. Thus, a test that is highly sensitive and specific will have a high PPV in a population with a disease that has a high prevalence and when the prevalence is low, as may be the case when the entire population is screened, then the PPV of the same test will be considerably lower. With a cut-off score of ≥7, the simplified FINDRISC had fairly acceptable sensitivity and specificity and, attained reasonable predictive values and diagnostic odds of detecting more individuals with undiagnosed diabetes than with the modified FINDRISC. However, with a cut-off of ≥ 8, both the modified FINDRISC and simplified FINDRISC had similar diagnostic accuracy as with a cut-off of ≥ 7 but with a lower sensitivity of 55.6% and 50.5% respectively. A cut-off score of ≥ 7 was therefore settled for, sacrificing a few PPV and diagnostic odds points but with a better sensitivity gain.

Using a simplified FINDRISC and the cut-off of 7 or more, 7.9% of those with a total score of ≥7 will be true positive while 98.7% of those whose total scores ≤ 7 will be true negatives. On the other hand, using a modified FINDRISC and its optimal cut-off of 7 or more, 6.9% will be true positives while 98.7% will be true negatives. For this study, using a simplified FINDRISC than the modified FINDRISC, and taking the same risk cut-off of 7 or more, the number of people identified for a laboratory test narrowed from 858 (21%) to 723 (18.0%) and the prevalence of undiagnosed diabetes increased from a pre-test probability of 1.8% to a post-test probability of 7.9% for simplified FINDRISC compared to 6.9% for modified FINDRISC. Considering that the prevalence of diabetes in the general population setting where this study analysis data was collected from is relatively low (about 2%), it may not necessarily be a waste of resources if such a risk score tool (with a PPV of 7.9% or 6.9%) is used for targeted screening to invite individuals with a score ≥7 for a more invasive diagnostic test particularly, at primary care setting in low-income countries where access to the diagnostic tests may not be readily available".

Whereas screening individuals for diabetes before its signs and symptoms develop is one approach thought to facilitate early diagnosis and initiation of effective lifestyle behavioral modifications and pharmacological interventions that can decrease diabetes disease progression especially among individuals with impaired glucose tolerance, it may not necessarily lead to reduced diabetes-related complication [53]. Some of the approaches to diabetes screening include: selective or targeted screening among population subgroups at relatively high risk regarding age, bodyweight, etc. and opportunistic screening carried out by health care professionals at a time when people are seen for different health reasons but with classic diabetes symptoms and risk factors e.g. obesity, high blood pressure etc. [54]. The added value of validated risk tools such as the one developed in this study is that the performance of the tool is

known. While at the individual level, clinicians might be fast taking BMI, waist circumference, and blood pressure measurements to assess risk profile, a standardized tool contribute to the quality-of-care, and allows population-based comparison. Furthermore, diabetes screening tests, especially in low-income countries include; risk assessment tools/questionnaires, biochemical tests such as; blood glucose measurement or glycated (HbA1c) measurement, and a combination of the two.

## Study strengths and limitations

The main strength of this study is the use of large national cross-sectional survey data. On the other hand, the study had limitations. First, this study did not evaluate the detection of future incident diabetes because the STEPwise cross-sectional survey was designed to collect data only at a particular point in time. However, FINDRISC was developed to detect undiagnosed diabetes and predict future incident diabetes [22]. Secondly, this study performed a single fasting plasma glucose (FPG) test to measure participants' blood glucose values as an outcome variable of interest, and could have underestimated the prevalence of diabetes in this study. Whereas OGTT and HbA1c tests are recommended to provide a more accurate clinical diagnosis of diabetes, the FPG test can equally provide a fairly accurate, and reliable population-level epidemiological diagnosis of diabetes in a given population [54], and we believe this study results are informative, and of added value to guide policy decision, and future research Thirdly, the use of self-reported physical activity and fruit and/or vegetable consumption data among other forms of self-reported data that may have attenuated this study's findings. There is also potential uncertainty surrounding the realization of participant fasting glucose state although study procedures required measurement of blood glucose after participant overnight fasting.

## Conclusion

The performance of the modified and simplified FINDRISC in detecting undiagnosed diabetes in Kenya was lower than the findings of the original FINDRISC but with an acceptable diagnostic AUC of 0.7486. With the modifications made to the original FINDRISC the tool can be usefully applied for a community screening of diabetes in the Kenyan population, and at a cutoff score of $\geq 7$, the simplified FINDRISC can reasonably discriminate individuals with diabetes from those without, making diabetes diagnosis more cost-effective.

## Supporting information

**S1 Fig. Finnish Diabetes Risk Score tool.**
(TIFF)

**S2 Fig. Modified Finnish Diabetes Risk Score tool.**
(TIFF)

**S1 Table. Sensitivity, specificity, diagnostic accuracy, PPV and NPV for different ages.**
(TIFF)

## Acknowledgments

We appreciate individuals and institutions that provided technical support for the design and implementation of the main survey. Authors further appreciate the data collection team led by the Ministry of Health (MoH) Kenya, WHO and Kenya National Bureau of Statistics (KNBS),

Kenya Medical Research Institute (KEMRI), and African Institute for Health and Development (AIHD).

## Author Contributions

**Conceptualization:** Innocent B. Mugume.

**Formal analysis:** Innocent B. Mugume, Solomon T. Wafula, Damazo T. Kadengye, Josefien Van Olmen.

**Methodology:** Innocent B. Mugume.

**Visualization:** Innocent B. Mugume, Solomon T. Wafula, Damazo T. Kadengye, Josefien Van Olmen.

**Writing – original draft:** Innocent B. Mugume.

**Writing – review & editing:** Innocent B. Mugume, Solomon T. Wafula, Damazo T. Kadengye, Josefien Van Olmen.

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
