## [Decision Letter · Decision Letter 0]

18 Mar 2022

PONE-D-21-37088The performance of a Finnish Diabetes Risk score (FINDRISC) in detecting undiagnosed diabetes among Kenyan adults aged 18-69 years.PLOS ONE

Dear Dr. Mugume

Thank you for submitting your manuscript to PLOS ONE. After careful consideration, we feel that it has merit but does not fully meet PLOS ONE’s publication criteria as it currently stands. Therefore, we invite you to submit a revised version of the manuscript that addresses the points raised during the review process.

We look forward to receiving your revised manuscript.

Kind regards,

Xianwu Cheng, M.D., Ph.D., FAHA

Academic Editor

PLOS ONE

Journal Requirements:

2. Please include a complete copy of PLOS’ questionnaire on inclusivity in global research in your revised manuscript. Our policy for research in this area aims to improve transparency in the reporting of research performed outside of researchers’ own country or community. The policy applies to researchers who have travelled to a different country to conduct research, research with Indigenous populations or their lands, and research on cultural artefacts. The questionnaire can also be requested at the journal’s discretion for any other submissions, even if these conditions are not met.  Please find more information on the policy and a link to download a blank copy of the questionnaire here: https://journals.plos.org/plosone/s/best-practices-in-research-reporting. Please upload a completed version of your questionnaire as Supporting Information when you resubmit your manuscript

Reviewers' comments:

Reviewer's Responses to Questions

**Comments to the Author**

1. Is the manuscript technically sound, and do the data support the conclusions?

Reviewer #1: Yes

Reviewer #2: No

2. Has the statistical analysis been performed appropriately and rigorously? 

Reviewer #1: Yes

Reviewer #2: Yes

3. Have the authors made all data underlying the findings in their manuscript fully available?

Reviewer #1: Yes

Reviewer #2: No

4. Is the manuscript presented in an intelligible fashion and written in standard English?

Reviewer #1: Yes

Reviewer #2: Yes

5. Review Comments to the Author

Reviewer #1: It is very interesting to use a risk score in pickup people at risk in low economic countries, the key is to evaluate a suitable existing engine tool and revise it to adopt in a sepecific population. You had done a valuable study, and found a simplifies risk score to cover most of the people at risk.

Reviewer #2: The theme of this paper is important and have huge potential benefit to the developing nations. The major issue is that the conclusion of this paper lack significant statistical support. I suggest the authors to significantly increase the sample size for statistically significant conclusion

6. PLOS authors have the option to publish the peer review history of their article (what does this mean?). If published, this will include your full peer review and any attached files.

Reviewer #1: **Yes: **Benli Su, Dept of Endocrinology and Metabolism, Second Hospital of Dalian Medical University, Dalian, China

Reviewer #2: No

---

## [Author Response · Author response to Decision Letter 0]

14 May 2022

Response to the reviewers' comments have been provided, and attached as a separate file along with the revised manuscript

---

## [Decision Letter · Decision Letter 1]

21 Jun 2022

PONE-D-21-37088R1Performance of a Finnish Diabetes Risk Score in detecting undiagnosed diabetes among Kenyans aged 18-69 years.PLOS ONE

Dear Dr. Mugume 

Thank you for submitting your manuscript to PLOS ONE. After careful consideration, we feel that it has merit but does not fully meet PLOS ONE’s publication criteria as it currently stands. Therefore, we invite you to submit a revised version of the manuscript that addresses the points raised during the review process. Please submit your revised manuscript by August 10, 2022. If you will need more time than this to complete your revisions, please reply to this message or contact the journal office at plosone@plos.org. Please include the following items when submitting your revised manuscript:A rebuttal letter that responds to each point raised by the academic editor and reviewer(s). You should upload this letter as a separate file labeled 'Response to Reviewers'.A marked-up copy of your manuscript that highlights changes made to the original version. You should upload this as a separate file labeled 'Revised Manuscript with Track Changes'.An unmarked version of your revised paper without tracked changes. You should upload this as a separate file labeled 'Manuscript'.

We look forward to receiving your revised manuscript.

Kind regards,

Xianwu Cheng, M.D., Ph.D., FAHA

Academic Editor

PLOS ONE

Additional Editor Comments (if provided):

The original reviewers have decliend to revew second peer-review.

Thus, this academc editor have recruited new additional reviewers.

Reviewers' comments:

Reviewer's Responses to Questions

**Comments to the Author**

1. If the authors have adequately addressed your comments raised in a previous round of review and you feel that this manuscript is now acceptable for publication, you may indicate that here to bypass the “Comments to the Author” section, enter your conflict of interest statement in the “Confidential to Editor” section, and submit your "Accept" recommendation.

Reviewer #3: (No Response)

Reviewer #4: All comments have been addressed

2. Is the manuscript technically sound, and do the data support the conclusions?

Reviewer #3: Partly

Reviewer #4: Yes

3. Has the statistical analysis been performed appropriately and rigorously? 

Reviewer #3: Yes

Reviewer #4: Yes

4. Have the authors made all data underlying the findings in their manuscript fully available?

Reviewer #3: Yes

Reviewer #4: Yes

5. Is the manuscript presented in an intelligible fashion and written in standard English?

Reviewer #3: Yes

Reviewer #4: Yes

6. Review Comments to the Author

Reviewer #3: General Comments

In this study, using a modified and simplified FINRISC to detect undiagnosed diabetes in 4027 Kenyans aged 18-69 years, the authors examined how correct this inference of being diabetic. They found that this screening was acceptable to Kenyans, as already reported in some countries. They concluded that the simplified FINDRISC can distinguish individuals in a more cost-effective way, with or without diabetes. Statistical analysis seems sound. The manuscript is well written.

I have the following concerns:

Specific comments

1.　Sensitivity, specificity, and diagnostic accuracy depend on the prevalence of diabetes and are also influenced by race and lifestyle. As the author discussed in the text, there are quite a few reports on the accuracy of FINRISC for diabetes screening, including reports from Africa. The results shown here are unattractive.

2.　The original Finnish diabetes risk score was applied to subjects aged 35-64 years who were not treated with anti-diabetic drugs and was prospectively followed up for ∼10 years on the onset of diabetes, but the subjects recruited in this study were 27 to 60 years old (median 35 years, younger than the original study, where subjects were getting older during the follow-up of ∼10 years). The method and subject were different from this study. The original study predicted new cases of diabetes during the follow-up of ∼10 years. The meaning is different from this study. The results obtained here cannot be compared to the original results.

3.　Even If this screening contributes to the detection of diabetes, is its use really useful? Determination of HbA1c and blood glucose levels is essential for the diagnosis of diabetes. If you want to diagnose current diabetes instead of predicting future diabetes, you should determine them. Those costs are not high.

4.　Line210: The number of subjects is wrong. 137, not 136.

Reviewer #4: The sample size question previously asked has been adequately answered.

You could say in the discussion that your sample size, while limited by the prior collection before this analysis was undertaken, is still comparable to those in other studies such as the original study in Finland and the one in Philippines.

I am a new reviewer and have not seen the paper before. I have no major concerns.

I do have some suggestions on improving the paper for the reader.

Introduction page 4 line 57 suggest change to "inconvenient"

In the first sentence of the last paragraph of the introduction add that FINDRISC was originally developed and used to predict risk of developing T2DM but has also been used to detect diabetes References 26,27,28.

Study design Suggest move "conducted between April and June 2015" to household survey "conducted between April and June 2015" AND designed to...

Definitions

Omitting the 2 hour OGTT glucose will miss some cases. That's inevitable in this analysis but should be noted in the limitations. See Diabetic Medicine 2000 17(10): 741-745

Criteria for BMI and waist circumference. Are these the ones appropriate for this ethnic group. Please specify.

Original Finnish (FINDRISC) diabetes risk score

A score of 21 or more predicts a 50% likelihood of developing T2DM in the next 10 years. What score do they use as a cut off for detecting DM which is what you are doing. Please include both these points here.

Modified FINDRISC It says personal history of diabetes was collected while in the original FINDRISK it was a history of a high glucose. You should note this here. The same applies for simplified FINDRISC. Also table 1 has high blood glucose history. It needs to be very clear and the text and table should be consistent.

Table 3 has LR and text has LHR.

Discussion Last paragraph suggest >>"contributeS to the quality of care standard..

Omitting the 2 hour OGTT glucose will miss some cases. That's inevitable in this analysis but should be noted in the limitations. See Diabetic Medicine 2000 17(10): 741-745

7. PLOS authors have the option to publish the peer review history of their article (what does this mean?). If published, this will include your full peer review and any attached files.

Reviewer #3: No

Reviewer #4: No

---

## [Author Response · Author response to Decision Letter 1]

5 Aug 2022

We appreciate the reviewers for the comments and concerns raised during the review process. We believe we have adequately responded to the raised concern, and have no specific comments to any reviewers.

---

## [Decision Letter · Decision Letter 2]

17 Oct 2022

Performance of a Finnish Diabetes Risk Score in detecting undiagnosed diabetes among Kenyans aged 18-69 years.

PONE-D-21-37088R2

Dear Dr. Mugume

We’re pleased to inform you that your manuscript has been judged scientifically suitable for publication and will be formally accepted for publication once it meets all outstanding technical requirements.

Kind regards,

Xianwu Cheng, M.D., Ph.D., FAHA

Academic Editor

PLOS ONE

Additional Editor Comments (optional):

Although two original reviewers (#2 and #3), the all concerns raised four reviewers have addressed by the authors two round peer-review processes.

Reviewers' comments:

Reviewer's Responses to Questions

**Comments to the Author**

1. If the authors have adequately addressed your comments raised in a previous round of review and you feel that this manuscript is now acceptable for publication, you may indicate that here to bypass the “Comments to the Author” section, enter your conflict of interest statement in the “Confidential to Editor” section, and submit your "Accept" recommendation.

Reviewer #4: All comments have been addressed

2. Is the manuscript technically sound, and do the data support the conclusions?

Reviewer #4: Yes

3. Has the statistical analysis been performed appropriately and rigorously? 

Reviewer #4: Yes

4. Have the authors made all data underlying the findings in their manuscript fully available?

Reviewer #4: Yes

5. Is the manuscript presented in an intelligible fashion and written in standard English?

Reviewer #4: Yes

6. Review Comments to the Author

Reviewer #4: Thank you for your responses and modifications to the text. The issues raised have been adequately addressed in the revision

7. PLOS authors have the option to publish the peer review history of their article (what does this mean?). If published, this will include your full peer review and any attached files.

Reviewer #4: No

---

## [Editor Report · Acceptance letter]

1 Nov 2022

PONE-D-21-37088R2 

Performance of a Finnish Diabetes Risk Score in detecting undiagnosed diabetes among Kenyans aged 18–69 years. 

Dear Dr. B. Mugume:

I'm pleased to inform you that your manuscript has been deemed suitable for publication in PLOS ONE. Congratulations! Your manuscript is now with our production department. 

Kind regards, 

on behalf of

Associate Prof. Xianwu Cheng 

Academic Editor

PLOS ONE